# Microfluidic Preparation of ^89^Zr-Radiolabelled Proteins by Flow Photochemistry

**DOI:** 10.3390/molecules26030764

**Published:** 2021-02-02

**Authors:** Daniel F. Earley, Amaury Guillou, Dion van der Born, Alex J. Poot, Jason P. Holland

**Affiliations:** 1Department of Chemistry, University of Zurich, Winterthurerstrasse 190, CH-8057 Zurich, Switzerland; daniel.earley@chem.uzh.ch (D.F.E.); amaury.guillou@chem.uzh.ch (A.G.); 2FutureChemistry Agro Business Park 10, 6708 PW Wageningen, The Netherlands; d.vanderborn@futurechemistry.com; 3Department of Radiology and Nuclear Medicine, UMC Utrecht, Heidelberglaan 100, 3584 CX Utrecht, The Netherlands; a.j.poot@umcutrecht.nl

**Keywords:** flow chemistry, photochemistry, radiochemistry, protein conjugation

## Abstract

^89^Zr-radiolabelled proteins functionalised with desferrioxamine B are a cornerstone of diagnostic positron emission tomography. In the clinical setting, ^89^Zr-labelled proteins are produced manually. Here, we explore the potential of using a microfluidic photochemical flow reactor to prepare ^89^Zr-radiolabelled proteins. The light-induced functionalisation and ^89^Zr-radiolabelling of human serum albumin ([^89^Zr]ZrDFO-PEG_3_-Et-azepin-HSA) was achieved by flow photochemistry with a decay-corrected radiochemical yield (RCY) of 31.2 ± 1.3% (*n* = 3) and radiochemical purity >90%. In comparison, a manual batch photoreactor synthesis produced the same radiotracer in a decay-corrected RCY of 59.6 ± 3.6% (*n* = 3) with an equivalent RCP > 90%. The results indicate that photoradiolabelling in flow is a feasible platform for the automated production of protein-based ^89^Zr-radiotracers, but further refinement of the apparatus and optimisation of the method are required before the flow process is competitive with manual reactions.

## 1. Introduction

Due to their unique structures and functions, high affinity and target specificity, protein-based drug-conjugates have fast become essential tools in medical imaging. Protein-derived immunoglobulin fragments and monoclonal antibodies (mAbs) are frequently utilised for the development of both therapeutic and diagnostic agents [1]. For example, in the clinic, mAbs functionalised with metal-binding chelates such as desferrioxamine B (DFO) and radiolabelled with zirconium-89 (^89^Zr-mAbs) provide sophisticated, rationally designed, radiopharmaceuticals for use in positron emission tomography (PET) [2].

The preparation of radiolabelled protein-conjugates requires the formation of a new covalent bond between the protein and the ligand, which must be achieved without disrupting the structural and biological properties of the target protein. Standard conjugation methods require the use of pre-installed reactive groups such as activated esters or benzyl-isothiocyanates, prior chemical modification and/or pre-activation of the protein [3], and rely on thermochemically-driven reactions (at room temperature to 37 ^°^C) with amino-acid side chains or glycans [4,5,6,7]. These processes often include time consuming multi-step syntheses that are difficult to automate, and can be incompatible with protein formulation buffers which mandate pre-purification of the protein vector. 

In the last decade, flow chemistry has been combined with radiochemistry to prepare a variety of radiotracers with different radionuclides, including ^11^C [8,9] and ^18^F [10,11,12,13]. Small reaction volumes, efficient mixing and reproducible control over all essential reaction parameters are some of the features that make microfluidic reactions attractive for radiochemistry. In 2016, Wright et al. [14] demonstrated the microfluidic radiolabelling of ^89^Zr-mAbs. In 2019, Poot et al. [15] reported proof-of-concept studies demonstrating that automated radiolabelling of ^89^Zr-mAbs in a batch reactor can also be combined with automated purification. However, both approaches relied on the use of pre-functionalised proteins bearing the DFO chelate. In contrast, we provide a proof-of-concept showing that flow-based photochemistry can be combined with radiochemistry to produce ^89^Zr-radiolabelled proteins direct from the unfunctionalised (native) protein source.

Light-induced functionalisation of proteins with metal-binding chelates bearing photochemically active groups presents an alternative to traditional protein conjugation chemistries [4,16]. The aryl azide (ArN_3_) group absorbs various wavelengths of light (302–400 nm) to generate highly reactive nitrenes [17], which can be utilised to form new covalent bonds to a target protein [16]. This light-induced process is compatible with biologically relevant media, and occurs extremely rapidly (lifetimes of reactive intermediates are in the nanosecond to microsecond range) compared with traditional bioconjugation methods that react directly with native functional groups on the protein [18]. The proposed reaction mechanism favours the formation of a seven-membered azepine ring species kinetically which can then react rapidly with nucleophiles like primary amines to form a new covalent bond [16].

We recently reported the photoradiosynthesis of several viable ^68^Ga^3+^ and ^89^Zr^4+^ protein-conjugate PET radiotracers, from photoactivatable metal-binding chelates functionalised with an ArN_3_ group [19,20,21,22]. Importantly, this photochemical conjugation process occurs at wavelengths that do not disrupt protein structure or function. Photoradiolabelling to produce ^89^Zr-mAbs is compatible with several different antibody formulations (with mixtures containing large quantities of histidine; ascorbic acid; sugars such as α,α-trehalose; and surfactants such as polysorbate 20) which allows for direct protein-conjugation without the need for pre-purification of the protein before performing the bioconjugation step.

To enhance the water-solubility of the photoactivatable DFO derivatives, we recently introduced two new compounds (including DFO-PEG_3_-Et-ArN_3_
**1**; Scheme 1) that link the metal binding chelate to the ArN_3_ group via a polar *tris*-polyethylene glycol (PEG_3_) linker [22]. Compound **1** is an excellent ligand for exploring the potential for automated radiosynthesis of ^89^Zr-radiolabelled protein conjugates via photochemistry in flow (Figure 1).

Here, we present the synthesis of a radiolabelled protein conjugate prepared by light-induced photoconjugation using a microfluidic photochemical reactor in continuous flow. [^89^Zr]ZrDFO-PEG_3_-Et-azepin-HSA was isolated after photoactivation of [^89^Zr]ZrDFO-PEG_3_-Et-ArN_3_ and conjugation to human serum albumin (HSA) in a custom microfluidic photochemical apparatus. The results provide confidence that an automated procedure can be developed for the radiosynthesis of ^89^Zr-labelled proteins for future applications in PET starting from the native unfunctionalised protein.

## 2. Results and Discussion

### 2.1. Synthesis of DFO-PEG_3_-Et-ArN_3_, *(**1**)*

The functionalised DFO-PEG_3_-Et-ArN_3_ metal binding chelate (compound **1**; Scheme 1) was synthesised with an overall yield of 24% [22]. Briefly, treatment of 3-(4-aminophenyl)propanoic acid with imidazole-1-sulphonyl azide HCl [23], potassium carbonate and a catalytic amount of copper(II) sulphate pentahydrate furnished the corresponding *para*-substituted aryl azide compound **2** with a 76% yield. In parallel, the preparation of the mono-protected PEG_3_ linker derivative **3** was achieved by treating 4,7,10-trioxa-1,13-tridecanediamine with di-*tert*-butyl dicarbonate [24]. Then, the amide coupling of the carboxylic acid functionalised aryl azide derivative **2** with the mono-protected amine **3** was accomplished by the treatment of *O*-(7-azabenzotriazol-1-yl)-*N*,*N*,*N″*,*N″*-tetramethyluronium hexafluorophosphate (HATU) and *N*,*N*-diisopropylethylamine (DIPEA) in anhydrous DMF. Purification by flash column chromatography afforded compound **4** (82% yield). Deprotection of compound **4** with trifluoroacetic acid (TFA) and subsequent treatment with succinic anhydride furnished compound **6** (via compound **5**; 66% yield over two steps). Finally, HATU-mediated coupling of compound **6** with DFO mesylate afforded the target compound DFO-PEG_3_-Et-ArN_3_ (**1**)—58% yield. Full experimental details and characterisation data, including ^1^H and ^13^C{^1^H} NMR spectroscopy and high-resolution electrospray ionisation mass spectrometry for compound **1** and associated intermediates, are provided in the Appendix A.

### 2.2. Chip Design and Instrumentation

Flow photoradiochemistry was performed by using a FutureChemistry FlowStart B-222 photochemistry module (Figure 2A) equipped with a twin light-emitting diode (LED) light source (365 nM; LedEngin, Inc.) connected in series (Figure 2B). Light intensity was set to 100% power and was controlled by using a prototype FutureChemistry B-271 photochemistry module. The emission profile was measured experimentally with an emission maximum observed at 366.5 nm (Figure 2C; full-width at half-maximum ≈14 nm). The photochemical flow reaction was performed using a mounted FutureChemistry borosilicate glass microfluidic chip with approximately 700 wide and 500 µm deep channels and a total internal volume of 112 µL (Figure 2D).

The chip is comprised of two segments including a shorter split and recombine mixing section (which splits the flow and recombines it multiple times to ensure homogeneity of solutions injected), and a longer linear reaction channel. Due to the small channels, the surface area of the microfluidic chip is much larger compared to standard batch reactors, which ensures efficient heat transfer and exposure of the reagents to light for sufficient time to complete the photochemical activation of the ArN_3_ group [25,26].

### 2.3. Flow Radiochemistry

The [^89^Zr]ZrDFO-PEG_3_-Et-azepin-HSA protein conjugate was prepared by flow photoradiochemistry on a microfluidic chip by reaction of a pre-labelled solution of [^89^Zr]ZrDFO-PEG_3_-Et-ArN_3_ (^89^Zr-**1**^+^; solution A) at a pH of 8.0 to 8.5, and a solution of native (unfunctionalised) HSA in Chelex-treated water (solution B; 45 mg mL^–1^ protein concentration; Scheme 2). As an example, solution A was prepared by incubating ligand **1** (20 μL; 2 mM stock solution <1% DMSO/H_2_O) with aliquots of [^89^Zr][Zr(C_2_O_4_)_4_]^4–^ (40 μL; 5.532 MBq) in H_2_O (40 μL; pre-treated with Chelex-100 resin) at room temperature and at a pH of 8.0 to 8.5 (the optimal range for ArN_3_ photoconjugation) [19,20]. Quantitative ^89^Zr-radiolabelling yields were obtained in <5 min and ^89^Zr-**1**^+^ was characterised by radio-instant thin layer chromatography (radio-iTLC; Figure 3A) and radio-HPLC methods (Figure 3B). The chemical identity and radiochemical purity (RCP) of ^89^Zr-**1**^+^ (Figure 3B; blue trace) was confirmed by comparison of the elution profile of the corresponding [^nat^Zr]ZrDFO-PEG_3_-Et-ArN_3_ (^nat^Zr-**1**^+^) complex (Figure 3B; green trace). The ^89^Zr-**1**^+^ complex was then used in the microfluidic photoconjugation reaction without further purification. Irradiation of a solution of ^nat^Zr-**1**^+^ with a powerful LED confirmed the complex was photochemically active (Figure 3B; green trace).

Then, by using a pair of syringe pumps operated by independent drive units, 100 µL of solution A and 100 µL of solution B were injected simultaneously onto the microfluidic chip at a flow rate of 5 µL/min. The reaction vessel was then irradiated at a wavelength of 365 nm for 20 min (100 μL total volume of chip). After this time, 150 µL of H_2_O (for each syringe) was injected on to the chip (to flush the system) with irradiation continuing for a further 20 min at the same flow rate. The crude product was collected in an Eppendorf tube, reaching a final volume of approximately 500 µL.

Aliquots of the crude [^89^Zr]ZrDFO-PEG_3_-Et-azepin-HSA protein conjugate mixtures were retained for analysis and fractions purified by size-exclusion gel filtration (PD-10) chromatography. Crude and purified samples were then characterised by radio-iTLC, analytical PD-10 size-exclusion chromatography (SEC), and automated radio-HPLC equipped with a SEC gel-filtration column (Figure 4). After optimisation of the reaction conditions, the decay-corrected radiochemical yield (RCY) for the isolated [^89^Zr]ZrDFO-PEG_3_-Et-azepin-HSA product was 31.2 ± 1.3% (*n* = 3 independent experiments; with final protein concentration in the reaction mixture of 135 μM; errors reported as 1 standard deviation). For each reaction the radiochemical purity (RCP) of the isolated product was >90% (determined by HPLC). The fraction of protein aggregation (indicated by an asterisk in Figure 4C) was <10%. Experimental data confirm that the ^89^Zr-radiolabelled proteins can be produced by photochemical methods in an automated, microfluidic system.

For comparison, manual reactions were also performed by using direct, top-down irradiation a stirred reaction mixture in a ≈1 mL glass vial. In this manual approach previously reported by Guillou et al., [22] [^89^Zr]ZrDFO-PEG_3_-Et-azepin-HSA was produced with a decay-corrected RCY of 59.6 ± 3.6% (*n* = 3) and with a RCP > 90%. Further refinement of the microfluidic apparatus and optimisation of the chemical methods is required before the process is competitive with manual reactions.

## 3. Conclusions

The synthesis of [^89^Zr]ZrDFO-PEG_3_-Et-azepin-HSA, was achieved by light-induced flow photoconjugation by using commercially available photochemistry modules. Using a microfluidic chip comprised of a split and recombine mixing section and a linear reaction section, coupled to a twin LED light source, [^89^Zr]ZrDFO-PEG_3_-Et-azepin-has was prepared in a radiochemical yield of 31.2 ± 1.3% (*n* = 3) in high radiochemical purity (>90%). The results provide an encouraging proof-of-concept that continuous flow procedures can be developed to produce protein-based radiotracers by automated instrumentation.

## 4. Experimental

### 4.1. General

All reagents and anhydrous solvents were purchased from commercial sources (Sigma-Aldrich (St. Louis, MO, USA), Merck (Darmstadt, Germany), Tokyo Chemical Industry (Eschborn, Germany) or abcr (Karlsruhe, Germany)) and were used without any further purification unless otherwise stated. All aqueous reactions were carried out using MilliQ H_2_O (>18.2 M MΩ·cm at 25 °C, Merck, Darmstadt, Germany). All anhydrous reactions were carried out in oven-dried glassware under an inert atmosphere. Reactions (where possible) were monitored using thin layer chromatography (TLC) analysis on aluminium plates coated with Silica Gel 60 F_254_ (E. Merck), and were visualised by short-wave ultra-violet irradiation (254 nm; where applicable), stained with ninhydrin in EtOH or propargyl alcohol and Cu(I)Br in EtOH, followed by charring at ≈200 °C. Purification was carried out by flash chromatography on a column of silica gel 60 (0.040–0.063 mm) or by reversed-phase C18 column chromatography using a Teledyne Isco CombiFlash^®^ Rf+ Lumen flash chromatography system fitted with RediSep Rf Gold^®^ reversed-phase C18 columns (5 to 50 g), eluting in a gradient of 0 to 100% of solvent B (MeOH with 0.1% TFA added). Solvent A: MilliQ H_2_O with 0.1% TFA added. Evaporation of solvents was performed under reduced pressure by using a rotary evaporator (Rotavapor R-300, Büchi Labortechnik AG, Flawil, Switzerland).

^1^H NMR and ^13^C{^1^H} NMR experiments were performed using deuterated solvents (Sigma-Aldrich, St. Louis, MO) on a Bruker AV-400 (^1^H: 400 MHz, ^13^C: 100.6 MHz) or a Bruker AV-500 (^1^H: 500 MHz, ^13^C: 125.8 MHz) spectrometer. Chemical shifts (δ) for ^1^H and ^13^C spectra are reported in parts per million (ppm) and are relative to the residual solvent peak. Coupling constants (*J*) are reported in Hz. Peak multiplicities are abbreviated as follows: s (singlet), d (doublet), dd (doublet of doublets), t (triplet), q (quartet), quint (quintet), m (multiplet), and br s (broaden singlet). Two-dimensional ^1^H-^1^H correlation spectroscopy (COSY) and ^13^C heteronuclear single quantum coherence (HSQC) NMR experiments were also performed to aid in the assignment of the ^1^H and ^13^C spectra, respectively. High-resolution electrospray ionisation mass spectrometry (HR-ESI-MS) was performed in either positive or negative ionisation mode (as indicated) using a Bruker MaXis QTOF-MS instrument (Bruker Daltronics GmbH, Bremen, Germany) and was measured by the mass spectrometry service at the Department of Chemistry, University of Zurich.

Analytical high-performance liquid chromatography (HPLC) experiments were performed using a Hitachi Chromaster Ultra Rs system fitted with a reversed-phase VP 250/4 Nucleodur C18 HTec (4 mm ID × 250 mm, 5 μm) column. This system was also fitted to a FlowStar^2^ LB 514 radioactivity detector (Berthold Technologies, Zug, Switzerland) equipped with a 20 μL PET cell (MX-20-6, Berthold Technologies) for analysing radiochemical reactions. Size-exclusion high-performance liquid chromatography (SEC-HPLC) experiments (for protein samples) were performed using a Rigol HPLC system (Contrec AG, Dietikon, Switzerland) equipped with an Enrich SEC 650 size-exclusion column (24 mL volume, 10 mm ID × 300 mm, Bio-Rad Laboratories, Basel, Switzerland). Electronic absorption was measured at 280 nm.

Purities of synthetic intermediates after chromatographic purification were judged to be >90% by analysis of ^1^H and ^13^C NMR spectra. Purities of final compounds were ≥95% (NMR or HPLC analysis), after reverse-phase C18 chromatography.

### 4.2. Synthesis of DFO-PEG_3_-Et-ArN_3_ *(**1**)*

**Synthesis of compound 2.** 3-(4-Aminophenyl)propanoic acid (1.00 g, 6.05 mmoL) was taken up in MeOH (30 mL) to which imidazole-1-sulphonyl azide HCl (1.52 g, 7.26 mmol), K_2_CO_3_ (2.26 g, 16.3 mmoL) and CuSO_4_·5H_2_O were added and the reaction mixture was stirred for 16 h at rt. The reaction was monitored by TLC and on complete conversion of the starting materials, the mixture was concentrated under reduced pressure. The resulting crude residue was dissolved in H_2_O (60 mL), acidified with concentrated HCl and extracted with EtOAc (3 × 50 mL). The organic layers were combined, dried over NaSO_4_ and concentrated under reduced pressure. The residue was then co-evaporated with cyclohexane to give compound **3** (875 mg, 76% yield) as a pale-yellow solid. **^1^H NMR** (400 MHz, CDCl_3_): δ 7.23–7.17 (m, 2H, CH_Ar_), 7.00–6.92 (m, 2H, CH_Ar_), 2.94 (t, J = 7.6 Hz, 2H, CH_2_), 2.67 (t, J = 7.7 Hz, 2H, CH_2_). **^13^C{^1^H} NMR** (101 MHz, CDCl_3_): δ 178.5 (C=O), 138.4 (C_qt_), 137.0 (C_qt_), 129.8 (CH_Ar_), 119.3 (CH_Ar_), 35.6 (CH_2_), 30.1 (CH_2_). **HR-ESI-MS** (negative mode): *m/z* calcd. for C_9_H_9_N_3_O_2_ [M − H]^−^ 190.0622, found 190.0621.

**Synthesis of compound 4.** Compound **2** (300 mg, 1.57 mmol) and *N*-[(dimethylamino)-1*H*-1,2,3-triazolo-[4,5-β]pyridin-1-ylmethylene]-*N*-methylmethanaminium hexafluorophosphate *N*-oxide (HATU; 895 mg, 2.36 mmoL) were taken up in dry DMF (4 mL) and stirred under N_2_ for 20 min at rt. To a solution of the *N*-Boc-4,7,10-trioxa-1,13-tridecanediamine, compound **3** (755 mg, 2.36 mmoL) dissolved in dry DMF (1 mL) was then added and the mixture stirred for a further 10 min. At this time, DIPEA (1.09 mL, 6.28 mmol) was added, and the reaction mixture was stirred at rt for 16 h under N_2_(g). The reaction was monitored by TLC, and on completion, the mixture was concentrated under reduced pressure. The crude residue was dissolved in EtOAc (25 mL) and washed with H_2_O (2 × 50 mL). The aqueous layers were then back-extracted with EtOAc (50 mL); the organic layers were combined and washed with brine, dried over NaSO_4_ and concentrated under reduced pressure. The crude residue was then purified by flash chromatography on a bed of silica [1/1 (*v*/*v*) EtOAc/hexane] to give compound **4** (638 mg, 82% yield) as a yellow to orange oil. **^1^H NMR** (400 MHz, CDCl_3_): δ 7.21–7.15 (m, 2H, CH_Ar_), 6.97–6.89 (m, 2H, CH_Ar_), 6.25 (s, 1H, NH), 4.92 (s, 1H, NH), 3.64–3.45 (m, 12H, CH_2_), 3.33 (q, *J* = 6.0 Hz, 2H, CH_2_), 3.19 (q, *J* = 6.3 Hz, 2H, CH_2_), 2.92 (dd, *J* = 8.5, 6.9 Hz, 2H, CH_2_), 2.46–2.37 (m, 2H, CH_2_), 1.75–1.69 (m, 4H, CH_2_), 1.42 (s, 9H, C(CH_3_)_3_). **^13^C{^1^H} NMR** (101 MHz, CDCl_3_): δ 171.9 (C=O), 156.2 (C=O), 138.1 (C_qt_), 138.0 (C_qt_), 129.9 (CH_Ar_), 119.1 (CH_Ar_), 79.1 (C(CH_3_)_3_), 70.6 (CH_2_), 70.6 (CH_2_), 70.3 (CH_2_), 70.1 (CH_2_), 69.6 (CH_2_), 38.6 (CH_2_), 38.4 (CH_2_), 38.1 (CH_2_), 31.2 (CH_2_), 29.8 (CH_2_), 28.9 (CH_2_), 28.6 (C(CH_3_)_3_). **HR-ESI-MS** (positive mode): *m/z* calcd. for C_24_H_39_N_5_O_6_ [M + Na]^+^ 516.2796, found 516.2792.

**Synthesis of compound 5.** Compound **4** (638 mg, 1.29 mmoL) was dissolved in CH_2_Cl_2_ (10 mL) and cooled to 0 °C, and then TFA (2 mL) was added drop-wise. The reaction mixture was then allowed to warm slowly to rt and then stirred for 1 h. At this time, TLC (10% MeOH in EtOAc) showed complete consumption of the starting material. The reaction mixture was then concentrated under reduced pressure and purified by flash column chromatography (C18, H_2_O to 100% MeOH) to give compound **5** as a yellow oil (387 mg, 76% yield). **^1^****H NMR** (400 MHz, CDCl_3_): δ 7.18 (d, *J* = 8.0 Hz, 2H, CH_Ar_), 6.93 (d, *J* = 8.0 Hz, 2H, CH_Ar_), 6.59 (m, *J* = 5.6 Hz, 1H, NH), 3.68–3.60 (m, 7H, CH_2_), 3.57–3.53 (m, 2H, CH_2_), 3.46 (t, *J* = 5.7 Hz, 2H, CH_2_), 3.28 (q, *J* = 6.4 Hz, 2H, CH_2_), 3.02 (t, *J* = 5.9 Hz, 2H, CH_2_), 2.96–2.87 (m, 3H, CH_2_), 2.45 (t, *J* = 7.7 Hz, 2H, CH_2_), 1.85 (quint, *J* = 5.9 Hz, 2H, CH_2_), 1.70 (quint, *J* = 6.2 Hz, 2H, CH_2_), 1.36–1.20 (m, 2H, NH_2_). **^13^C{^1^H} NMR** (101 MHz, CDCl_3_): δ 172.7 (C=O), 138.0 (C_qt_), 138.0 (C_qt_), 129.9 (CH_Ar_), 119.2 (CH_Ar_), 70.5 (CH_2_), 70.5 (CH_2_), 70.0 (CH_2_), 69.9 (CH_2_), 69.2 (CH_2_), 40.3 (CH_2_), 38.2 (CH_2_), 37.2 (CH_2_), 31.2 (CH_2_), 29.2 (CH_2_), 29.2 (CH_2_). **HR-ESI-MS** (positive mode): *m/z* calcd. for C_19_H_31_N_5_O_4_ [M + H]^+^ 394.2449, found 394.2453.

**Synthesis of compound 6.** Compound **5** (224 mg, 0.55 mmoL) was taken up in dry DMF (5 mL). Succinic anhydride (110 mg, 1.10 mmoL) was added and the reaction mixture was stirred for 16 h at rt. The reaction was monitored by TLC, and on completion, the solvent was removed under reduced pressure and the crude mixture was purified by flash column chromatography (C18, H_2_O to 100% MeOH) to give compound **6** (235 mg, 87% yield) as an orange to brown oil. **^1^****H NMR** (400 MHz, MeOD): δ 7.24 (d, *J* = 8.0 Hz, 2H, CH_Ar_), 6.98 (d, *J* = 8.0 Hz, 2H, CH_Ar_), 3.66–3.47 (m, 11H, CH_2_), 3.39 (t, *J* = 6.2 Hz, 2H, CH_2_), 3.23 (dt, *J* = 19.5, 6.7 Hz, 4H, CH_2_), 2.90 (t, *J* = 7.5 Hz, 2H, CH_2_), 2.58 (t, *J* = 7.0 Hz, 2H, CH_2_), 2.50–2.42 (m, 4H, CH_2_), 1.71 (dquint, *J* = 30.2, 6.4 Hz, 4H, CH_2_). **^13^C{^1^H} NMR** (101 MHz, MeOD): δ 176.2 (C=O), 174.9 (C=O), 174.4 (C=O), 139.4 (CH_qt_), 139.2 (CH_qt_), 131.0 (CH_Ar_), 120.0 (CH_Ar_), 71.5 (CH_2_), 71.2 (CH_2_), 69.8 (CH_2_), 69.7 (CH_2_), 38.8 (CH_2_), 37.8 (CH_2_), 37.7 (CH_2_), 32.2 (CH_2_), 31.6 (CH_2_), 30.4 (CH_2_), 30.3 (CH_2_). **HR-ESI-MS** (positive mode): *m/z* calcd. for C_23_H_35_N_5_O_7_ [M+H]^+^ 494.2609, found 494.2611.

**Synthesis of DFO-PEG_3_-Et-ArN_3_ (1).** Compound **6** (222 mg, 0.45 mmoL) and HATU (232 mg, 0.61 mmoL) were dissolved in dry DMF (6 mL) and stirred for 20 min under N_2_(g). DFO mesylate (267 mg, 0.41 mmoL) was then dissolved in DMF (4 mL) and added to the reaction mixture with stirring for a further 10 min. At this time, DIPEA (0.29 mL, 1.63 mmoL) was added and the was reaction stirred under N_2_(g) for 16 h at rt. The reaction was monitored by TLC, and on completion the mixture was concentrated under reduced pressure and the crude residue was purified by flash column chromatography (C18, H_2_O/MeOH 0% MeOH to 100%) followed by washing with ice-cold acetone (6 × 5 mL; separated by centrifugation between each wash) to give compound **1** (DFO-PEG_3_-Et-ArN_3_; 244 mg, 58% yield) as an off-white solid. **^1^H NMR** (500 MHz, DMSO-*d*^6^): δ 9.62 (m, 2H, OH), 7.77 (q, *J* = 5.0, 4.6 Hz, 4H, NH), 7.23 (d, *J* = 8.4 Hz, 2H, CH_Ar_), 7.02 (d, *J* = 8.4 Hz, 2H, CH_Ar_), 3.57–3.42 (m, 14H, CH_2_), 3.37 (t, *J* = 6.4 Hz, 2H, CH_2_), 3.32 (t, *J* = 6.4 Hz, 2H, CH_2_), 3.10–2.94 (m, 10H, CH_2_), 2.79 (t, *J* = 7.6 Hz, 2H, CH_2_), 2.57 (t, *J* = 7.4 Hz, 4H, CH_2_), 2.33 (t, *J* = 7.7 Hz, 3H, CH_2_), 2.29–2.23 (m, 7H, CH_2_), 1.96 (s, 3H, CH_3_), 1.58 (dt, *J* = 13.7, 6.8 Hz, 4H, CH_2_), 1.49 (t, *J* = 7.2 Hz, 5H, CH_2_), 1.41–1.34 (m, 5H, CH_2_), 1.24–1.17 (m, 5H, CH_2_). **^13^C{^1^H} NMR** (126 MHz, DMSO-*d*^6^): δ 172.0 (C=O), 171.3 (C=O), 171.2 (C=O), 171.1 (C=O), 171.0 (C=O), 170.1 (C=O), 162.3 (C=O), 158.6–157.7 (TFA), 138.4 (CH_qt_), 136.9 (CH_qt_), 129.8 (CH_Ar_), 118.9 (CH_Ar_), 69.7 (CH_2_), 69.5 (CH_2_), 68.0 (CH_2_), 68.0 (CH_2_), 47.1 (CH_2_), 46.8 (CH_2_), 38.4 (CH_2_), 38.4 (CH_2_), 36.9 (CH_2_), 35.8 (CH_2_), 35.7 (CH_2_), 30.9 (CH_2_), 30.9 (CH_2_), 30.4 (CH_2_), 29.9 (CH_2_), 29.4 (CH_2_), 29.3 (CH_2_), 28.8 (CH_2_), 27.6 (CH_2_), 26.0 (CH_2_), 23.5 (CH_2_), 20.3 (CH_3_). **HR-ESI-MS** (positive mode): *m/z* calcd. for C_48_H_81_N_11_O_14_ [M + H]^2+^ 518.8055, found 518.8057.

### 4.3. Flow Photochemistry

Flow photoradiochemistry was performed using a FlowStart B-222 (Future Chemistry, Nijmegen, The Netherlands) photochemistry module equipped with a twin light-emitting diode (LED; LedEngin Inc., San Jose, CA, USA) light source (365 nM), connected in series. Light intensity was set to 100% power and controlled using a prototype Future Chemistry B-271 (Future Chemistry, Nijmegen, The Netherlands) photochemistry module. LED intensity was measured by using a S470C Thermal Power Sensor Head Volume Absorber, 0.25–10.6 µm, 0.1 mW–5 W, Ø15 mm. Light intensity for each LED was 366.5 nm (FWHM of ≈10 nm). The photochemical flow reactions were performed using a mounted Micronit microfluidics E3 custom borosilicate glass chip (Future Chemistry, Nijmegen, The Netherlands) with an internal diameter width of less than 700 µm, depth of 500 µm and a 112 µL total volume. The temperature of all photochemical conjugation reactions was typically 23 ± 2 °C (ambient conditions).

### 4.4. Radioactivity and Radioactive Measurements

All instruments for measuring radioactivity were calibrated and maintained in accordance with previously reported routine quality control procedures. [^89^Zr][Zr(C_2_O_4_)_4_]^4−^ was obtained as a solution in ≈1.0 M aq oxalic acid from PerkinElmer (Boston, MA, manufactured by the BV Cyclotron VU, Amsterdam, The Netherlands) and was used without further purification. Radioactive reactions were monitored by using instant thin-layer chromatography (radio-iTLC). Glass-fibre iTLC plates impregnated with silica-gel (iTLC-SG, Agilent Technologies) were developed in using aqueous mobile phases containing DTPA (50 mM, pH7.1) and were analysed on a radio-TLC detector (SCAN-RAM, LabLogic Systems Ltd, Sheffield, United Kingdom). Radiochemical conversion (RCC) was determined by integrating the data obtained by the radio-TLC plate reader and determining both the percentage of radiolabelled product (*R*_f_ = 0.0) and ‘free’ ^89^Zr (*R*_f_ = 1.0; present in the analyses as [^89^Zr][Zr(DTPA)]^−^). Integration and data analysis were performed by using the software Laura version 5.0.4.29 (LabLogic). Appropriate background and decay corrections were applied as necessary. Radiochemical purities (RCPs) of labelled protein samples were determined by size-exclusion chromatography (SEC) using two different columns and techniques. The first technique used an automated size-exclusion column (Bio-Rad Laboratories, ENrich SEC 70, 10 ± 2 µm, 10 mm ID × 300 mm) connected to a Rigol HPLC system (Contrec AG, Dietikon, Switzerland) equipped with a UV/visible detector (absorption measured at 220, 254 and 280 nm) and a radioactivity detector (FlowStar^2^ LB 514, Berthold Technologies, Zug, Switzerland). Isocratic elution with phosphate buffered saline (PBS, pH7.4) was used. The second method used a manual procedure involving size-exclusion column chromatography and a PD-10 desalting column (Sephadex G-25 resin, 85–260 µm, 14.5 mm ID × 50 mm, >30 kDa, GE Healthcare). For analytical procedures, PD-10 columns were eluted with PBS. A total of 40 × 200 µL fractions were collected up to a final elution volume of 8 mL. Note that the loading/dead-volume of the PD-10 columns was precisely 2.5 mL, which was discarded prior to aliquot collection. For quantification of radioactivity, each fraction was measured on a gamma counter (HIDEX Automatic Gamma Counter, Hidex AMG, Turku, Finland) using an energy window between 480 and 558 keV for ^89^Zr (511 keV emission) and a counting time of 30 s. Appropriate background and decay corrections were applied throughout. PD-10 SEC columns were also used for preparative purification and reformulation of radiolabelled products (in sterile PBS; pH 7.4) by collecting a fraction of the eluate corresponding to the high molecular weight protein (>30 kDa fraction eluted in the range 0.0 to 1.6 mL as indicated for each experiment).

### 4.5. ^89^Zr-Radioactive Stocks

Stock solutions of [^89^Zr][Zr(C_2_O_4_)_4_]^4–^ were prepared on several occasions using the same procedure. As an example, a stock solution of [^89^Zr][Zr(C_2_O_4_)_4_]^4−^ was prepared by adding ^89^Zr radioactivity from the source (89.11 MBq, 150 μL in ≈1.0 M aq oxalic acid; PerkinElmer) to an Eppendorf tube. The solution was neutralized by the addition of aliquots of 1.0 M aq Na_2_CO_3_ (total volume of 180 μL added, final pH ≈ 7.5–7.7, final volume ≈345 μL, final activity = 82.98 MBq). Caution: Acid neutralization with Na_2_CO_3_ releases CO_2_(g) and care should be taken to ensure that no radioactivity escapes the microcentrifuge tube. After CO_2_ evolution ceased, several different reactions were performed by using the same stock solutions.

## Data Availability

The data presented in this study are available on request from the corresponding author. All relevant data are presented in the manuscript and Appendix A.

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
