# Peer review of "Microfluidic Preparation of 89Zr-Radiolabelled Proteins by Flow Photochemistry"

_molecules, 2021, doi:10.3390/molecules26030764_

Round 1

Reviewer 1 Report

The manuscript titled "Microfluidic preparation of 89Zr-radiolabelled proteins by flow photochemistry" written by J. P. Holland and coworkers reported photochemical conjugation of HSA and 89Zr-labelled ArN3 in a micro-flow reactor. The Ar azide compound was prepared by a multi-step synthesis. The reported synthetic protocols and spectral data are reliable. The authors tried the conjugation using a commercially available photoreactor (FutureChemistry B-222) and obtained a desired conjugate in ca. 31% RCY. Unfortunately, the observed yields were lower than those under batch photoreactions (ca. 60% RCY). Although the reported results were not positive, I admit that this manuscript contains some valuable information. However, I think revisions of the manuscript is required before further considerations.

In the first paragraph, no previous reports were cited. Some representative reviews should be cited at least.

The authors did not describe why they planned to use micro-flow photoreactor in their research. The authors should cite reviews for photochemical flow reactions and should describe merits of using photochemical flow reactors (Photochemical micro-flow reactors have thin reaction channels, thus undesired photo decay can be suppressed based on Lambert-Beer low).

I could not speculate why flow conditions afforded worse results compared with those under batch conditions from the manuscript. Although positive results were not obtained unfortunately, considerations for the obtained results are important. The authors should describe detailed reaction conditions for batch photochemical reactions (information for light source, distance between the batch reactor and the light source, irradiation time, temperature control method is required).

Compound number "3" should be appeared in legend of Scheme 1.

Reviewer 2 Report

The manuscript submitted by Holland and co-workers describes in a very carefull and detailed way the radiolabelling of a model protein (human serum albumin) with the PET radionuclide zirconium-89 using a microfluidic photochemical flow reactor. The approach proposed is original and innovative, which merits publication in Molecules. The manuscript should be revised taking the following points into consideration:

  • Typos such as "Synthesised" should be corrected;

  • The synthesis of PEG3 linker derivative 3 should be either included in Scheme 1. or, alternatively, the caption of Scheme 1 could be modified as follows: "Scheme 1. ...(b) N-Boc-4,7,10-trioxa-1,13 tridecanediamine (3), HATU, DIPEA, anhyd. DMF, rt, 24 h;..."

  • In Figure 3B. the "purple" elution profile is missing; The sentence in lines 139/140 should be also revised accordingly.

Round 2

Reviewer 1 Report

I was satisfied with authors' revision. Now I can support this manuscript for publication in Molecules.